# A New Method of Preparing Aurone by Marine Actinomycetes and Its Potential Application in Agricultural Fungicides

**DOI:** 10.3390/molecules28010017

**Published:** 2022-12-20

**Authors:** Bin Liu, Xiaomeng Li, Weiguo Wang, Xin Wang, Pahaiding Aihaiti, Tingtang Lin, Zishuo Fu, Rui Xu, Mengqi Wu, Zhong Li, Yang Zhang

**Affiliations:** 1School of Pharmaceutical, East China University of Science and Technology, Shanghai 200237, China; 2Shanghai Qingpu District Agricultural Technology Extension Service Center, Shanghai 201700, China; 3School of Bioengineering, East China University of Science and Technology, Shanghai 200237, China

**Keywords:** aurone, marine actinomycetes, fermentation, plant pathogen, fungicidal activity

## Abstract

A strain of marine actinomycetes was isolated from an intertidal zone and identified as *Streptomyces cinereoruber*. Through the fermentation of this strain, a compound with fungicidal activity was extracted and purified. Using mass spectrometry (MS) and nuclear magnetic resonance (NMR) data, the metabolite was determined to be an aurone. The toxicity of the aurone toward four kinds of tumor cells—SH-SY5Y, HepG2, A549, and HeLa cells—was verified by the MTT method, delivering IC_50_ values of 41.81, 47.19, 63.95, and 51.92 μg/mL, respectively. Greenhouse bioassay showed that the aurone exhibited a high fungicidal activity against powder mildew (*Botrytis cinerea*), cucurbits powder mildew (*Sphaerotheca fuliginea* (Schlecht ex Ff.) Poll), and rice blast (*Pyricularia oryzae*).

## 1. Introduction

Aurones are naturally occurring yellow pigments of plants that are structurally related to flavonoids and have a broad spectrum of bioactivities, such as anticancer, antiviral, antiparasitic, antifungal, and antidiabetic activities [1,2]. Aurones rarely exist in plants and are generally artificially synthesized [1,2,3]. Notably, chloroaurone was isolated from a marine source, *Spatoglossum variabile* (brown alga) [4]. Although numerous land microorganisms have not been identified, finding new compounds or species from land microbes is becoming increasingly difficult, causing researchers to search for gems in marine microorganisms. As marine and land microorganisms are significantly different because of their environments, many new bioactive metabolites have been isolated from marine bacteria and fungi in recent decades [5,6,7]. Furthermore, there are many advantages in using microorganisms to synthesize compounds with bioactivities. Compared with organic synthesis technology, biosynthesis requires mild operating conditions. Furthermore, the products from biosynthesis retain their stereostructure. Compared with plant extraction, fermentation technology is more beneficial for the industrial manufacture of metabolites because the metabolite yield can be dramatically increased by mutagenic strains and by optimizing the fermentation conditions [8].

While screening agriculture-used bioactive metabolites from microorganisms, the fermentation broth of a marine actinomycete exhibited high activity against powder mildew (*Botrytis cinerea*), cucurbits powder mildew (*Sphaerotheca fuliginea* (Schlecht ex Ff.) Poll), and rice blast (*Pyricularia oryzae*). Thus, the active component was isolated and purified by solvent extraction, silica chromatography, and preparative high-performance liquid chromatography, and it was identified by mass spectrometry (MS) and nuclear magnetic resonance (NMR). Herein, the taxonomy of the marine actinomycete, a new method for preparing aurones from this strain, its cytotoxicity, and its potential application in agricultural fungicides are reported.

## 2. Results

### 2.1. Taxonomic Status of Strains

The morphological, physiological, and biochemical features of strain TX6 are described in Table 1 and Table 2.

The sequence of 16S rDNA is as described in Genbank No. HQ315846. According to the morphological, physiological, and biochemical features, along with the sequence of 16S rDNA, the strain was classified as *Streptomyces cinereoruber*.

### 2.2. Structural Elucidation of the Active Compound

The active compound was yellow amorphous powder. TOF-Mass (Figure 1) shows that the molecular formula of the active compound was C_15_H_10_O_4_ and *m*/*z* 255.2408 [m + H]+ (calcd for C_15_H_10_O_4_, 254.2408). The 1H NMR of the active compound (Appendix A) showed the existence of two hydroxyl proton signals at δH 11.13 (^1^H, s) and δH 10.15 (^1^H, s). The results of other ^1^H-NMR chemical shifts of this compound are shown in Table 3, revealing spectra (600 MHz in d6-DMSO, δ ppm)—6.78 (^1^H, d, *J* = 2.0 Hz, H-7), 6.70 (dd, *J* = 8.4, 2.0 Hz, H-5), 7.79–7.84 (^2^H, m, H-2′, 6′), 7.60 (^1^H, d, *J* = 8.4 Hz, H-4), 6.88–6.91 (^2^H, m, H-3′, 5′), and 6.73 (H, s, H-2″)—which are comparable with those reported of the 6, 4′-dihydroxyaurone standard, ^1^H-NMR (500 MHz in MeOD, δ ppm): 8.09 (^1^H, s, H-7), 7.98 (^1^H, d, *J* = 1.08 Hz, H-5), 7.35 (^2^H, d, *J* = 1.28 Hz, H-2′, 6′), 6.88 (^1^H, d, *J* = 0.46 Hz, H-4), 6.84 (^2^H, d, *J* = 1.38 Hz, H-3′, 5′), and 6.72 (H, s, H-2″). In addition, ^13^C NMR, DEPT, COSY, and HMBC spectra of the active materials were analyzed (Appendix A). The correlation between 1H-1H COSY and HMBC assigned the characteristic signals of carbon (Figure 1). It can be seen from the carbon spectrum that the characteristic peak carbonyl carbon appeared at the low field of about 180 ppm. The two carbon peaks on the olefin also appeared in the spectrum according to the rule of 100–165 ppm, in which the saturated olefin carbon moved to the low field area, about 145 ppm. Because the carbon on both sides of the phenolic hydroxyl group on the right benzene ring had the same effect, two higher carbon peaks appeared in the carbon spectrum. Thus, the chemical structure of the active compound was determined to be 6, 4′-dihydroxyaurone.

Hydrogen nuclear magnetic resonance showed that the compound has eight hydrogens connected to the carbons, which are located as follows, respectively: ^1^H NMR (500 MHz, MeOD): 8.09 (H, s, H-7), 7.98, (H, d, *J* = 1.08, H-5), 7.35 (^2^H, d, *J* = 1.28 Hz, H-2′,6′), 6.88 (H, d, *J* = 0.46 Hz, H-4), 6.84 (^2^H, d, *J* = 1.38 Hz, H-3′,5′), and 6.72 (H, s, H-2″). Thus, the active compound was identified as aurone, or benzalcoumaranone (Table 3).

### 2.3. Optimal Fermentation Medium and Process

The single-factor analysis showed that glucose and peanut powder are the favorite carbon and nitrogen sources for TXC-6 (Table 4). An optimized fermentation medium was conducted by the orthogonal experiments, as shown in Table 5 and Table 6.

Through the analysis of the corrected range R′ value in the orthogonal test, glucose was found to have the greatest impact on the fermentation in the four components, followed by peanut powder and starch, while sodium chloride has little impact. The importance of each factor and the relationship between the yields and their relative concentration of each factor could be analyzed intuitively by showing the mean value of four factors in different proportions as a line diagram by the software, as shown in Figure 2. Figure 2 indicates that glucose plays the most important role in the production of aurone. The yield increased significantly with the increase of glucose; however, a too-high concentration may produce sugar inhibition and affect the fermentation.

Thus, an improved fermentation medium composed of 4% glucose, 3% peanut powder, 2% starch, 0.5% yeast extract powder, 0.5% sodium chloride, and 0.005% dipotassium hydrogen phosphate was obtained. Compared with the original screening medium for actinomycetes, the optimal medium increased the fermentation yield by nearly 170%, to a highest level of 152.20 mg/L.

According to the one-way ANOVA analysis, the accumulation of metabolites was best when the initial pH was adjusted to 7.0 with significant difference from the others, as Figure 3A illustrates. Figure 3B demonstrates that 28 °C was the best culture temperature for the production with significant difference from the others. For the culture time, although the yield was highest when the fermentation period was 5 days, it did not reach a significant difference from that of 6 days, as Figure 3C illustrates. Considering the time and comprehensive cost, we think that the best fermentation period is 5 days. With the gradual increase in the rotating speed of the shaking, the output also increased gradually (Figure 3D). When the speed was set as 200 rpm, the relative yield was the highest at 117%, but there was no significant difference with 113% when the speed was set at 160 rpm. When it was further increased to 240 rpm, the fermentation yield decreased sharply. This may be related to the high shearing force received by the mycelium due to the high rotation speed.

### 2.4. Cytotoxicity to Four Kinds of Canner Cells

The toxicity of aurone to the activities of four kinds of tumor cells—SH-SY5Y cells, HepG2 cells, A549 cells, and HeLa cells—was verified by the MTT method (Figure 4). The results showed that aurones had inhibition to the four cells, with the IC_50_ of 41.81, 47.19, 63.95, and 51.92 μg/mL, respectively.

### 2.5. Fungicidal Activity In Vivo

The greenhouse pot experiment showed that the metabolites of bovis TX6 had good therapeutic effects on the diseases caused by *P. orizae*, *B. cinerea*, and *S. fulginea* at the concentration of 50 mg/L. At the concentration of 100 mg/L, the therapeutic effects were equivalent to that of the recommended dose for commercial products (Table 7).

The letters a, b, c, d, and e indicate the significant differences between different treatments in the same test, and the same letter indicates no significant difference.

## 3. Discussion

Compared with land organisms, knowledge of marine organisms is limited; specifically, the understanding of ocean microorganisms is considered to be superficial, at best. Understanding marine microorganisms can enable us to understand the evolutionary history of organisms from ocean to land and to fully utilize the diversity in microbial metabolism resulting from the differences between marine and terrestrial environments. This would enable us to obtain more compounds with new uses or mechanisms, which are beneficial for drug development. Aurones, found mainly in flowers of the *Scrophulariaceae* and *Compositae* families, are isolated and purified using an alcohol extraction mixture [9]. In addition to flowering plants, gymnosperms, bryophytes, and brown algae have been reported to be natural sources of aurones [10]. In this study, we initially found the marine actinomycete, TXC6, which was classified as *S. Cinereoruber* and could produce aurones. Aurones confer a bright golden color and fluorescence to flowers and plants and are produced mainly by the oxidative cyclization of 2′-hydroxychalcones in the presence of an enzyme, aurone synthase, in plants [11]. Interestingly, a strain of marine actinomycetes can produce these chemicals as secondary metabolites. Arguably, the peanut powder in the fermenting medium is the source of this aurone. However, from the medium without inoculated *S. cinereoruber*, TXC6, only trace amounts of isoflavone were found, and when nitrates were used as the alternative nitrogen resource, aurones were found in the fermented broth. Therefore, in the future, the mechanism of producing aurones by this strain needs to be analyzed using its relative gene.

One of the advantages of using microbial metabolites as pesticides is that the yield of metabolites can be significantly improved through strain mutation and the optimization of the fermentation conditions to promote industrialization [12,13]. All of the drugs from microbial fermentation metabolites, including pesticides, have experienced such a process. The production of many substances from microbial fermentation metabolites, such as drugs and pesticides, involves such improvement processes. The original *Penicillium notatum* supplied less than 2 U/mL of penicillin; however, the yield has now reached approximately 100,000 IU/mL through continuous mutation and process optimization. The development processes for Validamycin A [14], Avermectin [15], Spinosad [16], and other fermentation products in pesticides have experienced significant strain improvement and fermentation-process optimization. Therefore, improving the fermentation yield is essential for obtaining fermentation products. In our study, the preliminary optimization of the fermentation conditions was carried out to improve the fermentation yield of the strain to a certain extent; however, the degree of improvement was insufficient for industrialization, and more studies are required.

Powder mildew [17], cucurbits powder mildew [18], rice blast, and other plant fungal diseases [19] impair the production of related crops and decrease crop quality. Therefore, the development of new pesticides that can control these diseases has always been a research hotspot. As people become more concerned about their health, synthetic pesticide residues in agricultural products will receive more criticism; therefore, researchers are paying more attention to natural products in the research and development of new pesticides. This study found that the metabolites produced by TXC6 exert a good controlling effect on these diseases and have good developmental prospects. However, tracking the activity of isolated and purified compounds showed that with the continuous improvement in the purity of active compounds, their relative biological activity does not improve accordingly. This indicates that there may be multicomponent synergistic effects between the metabolites. The synergistic effect of multiple components is not only helpful in improving the fungicidal effect, but is also of great significance in delaying the occurrence of resistance. Therefore, the whole or crude extract of the fermentation broth is preferred for use as a pesticide, rather than a single compound after purification.

## 4. Material and Methods

### 4.1. Microorganism Material and Taxonomy

*Streptomyces cinereoruber* TXC6 (preserved in the China General Microbiological Culture Collection Center, CGMCC. No. 6922) was isolated from the intertidal zone in Dongfang, Hainan province, China. Microbiological classification methods were described as in previous documents [1].

### 4.2. Optimal Fermentation Experimental Design

The carbon and nitrogen sources in the fermentation medium were preliminarily selected by single-factor analysis, and a L9(3^4^) orthogonal design leading to 9 sets of experiments, performed in flasks with triplicates, was used to determine the optimal level of selected carbon sources, nitrogen sources, and inorganic salt composition.

Based on the optimal medium, further optimal fermentation-process experiments were carried out on initial pH, culture temperature, and fermentation time by single- factor tests in flasks with four replicates. For initial pH tests, the initial pH of the medium was adjusted to 5.0, 6.0, 7.0, 8.0, and 9.0, respectively, before sterilization. The sterilized media were inoculated and cultured at 28 °C for 4 days. For culture temperature tests, the culture temperatures were carried on at 22 °C, 25 °C, 28 °C, 31 °C, and 34 °C separately and cultured for 4 days. The harvest times were set as 3, 4, 5, 6, and 7 days for fermentation tests and shaker-rotate speeds of 80 rpm, 120 rpm, 160 rpm, 200 rpm, and 240 rpm were employed separately.

For routine analysis of aurone in the optimal fermentation process, the ferment broth was centrifuged and the supernatant was filtered by a 0.45 μm filter membrane for quantitative analysis by HPLC on an Anthena C18 column (250 mm × 4.6 mm, 10 μm, Anpel, Shanghai) at 25 °C with a flow rate of 1.0 mL/min by Anglent 1200 at 210 nm. The mobile phase consisted of 80% methanol and 20% water.

### 4.3. Culture, Extraction and Isolation

The isolated *S.cinereoruber* TXC6 was cultivated on the NA medium for about 10 days. The stainless-steel fermentation tank (B BRAUN, Braunschweig, Germany) with 10 L sterilized medium (4% glucose, 2% soluble starch, 3% peanut powder, 0.4% yeast extra, 0.1% beef meet, 0.005% K_2_HPO_4_, 0.5% NaCl, 0.1% MgCl_2_, 0.1% KCl, and 0.001% FeSO_4_) was inoculated with the matured clone. After 4 days of aerobic cultivation with agitation at 28 °C, the broth was filtered. The 10 L filtered aqua was extracted three times with water-saturated n-butanol. The organic phase was merged and evaporated in a vacuum to obtain the crude extract of about 15 g ointment. The ointment was subjected to chromatographing on silica gel, using gradients of dichloromethane initially, then gradients of dichloromethane with up to 1.67% of methanol. On the basis of TLC analysis, similar fractions were combined and were re-chromatographed on silica gel to yield 165 mg of active compound. Further purification was launched by preparative HPLC to obtain the active compound with the purity of 98%.

### 4.4. Mass and NMR

The high-resolution EI and FAB-mass spectra were recorded on a TOF mass spectrometer. Spectroscopic analyses were performed to determine the structure of the active substance. Nuclear magnetic resonance (NMR) spectra were recorded on a Bruker AMX500 MHz NMR spectrometer (Billerica, MA, USA). Spectra of 1H NMR were measured in deuterated methonal (MeOD) at room temperature.

### 4.5. Evaluation of the Cytotoxicity and Fungicidal Activity

#### 4.5.1. Cytotoxicity

MTT assay was used to determine cell viability and proliferation through cell metabolism levels [20,21]. The toxicity of aurone to the activity of four kinds of tumor cells was verified by the MTT method. SH-SY5Y cells, HepG2 cells, A549 cells, and HeLa cells were made into a single cell suspension of less than 1 × 10^5^ cells/mL. A 100 µL suspension was spread in a sterile 96-well plate and continuously cultured for 24 h, then treated with different concentrations of aurone for 24 h. After 24 h, 20 µL of 5 mg/mL MTT reagent was added to the 96-well plate and incubated for 4 h. Then, the medium containing thiazolyl blue (MTT) was removed, and 150 µL of DMSO was added. Finally, the absorbance (OD) value was determined under the single-wavelength condition (572 nm) with a full-function microplate reader. The inhibition rate (%) equaled [1 − drug treatment group (OD_λ572_)/blank control group (OD_λ572_)] × 100%.

#### 4.5.2. Bioassay of Powder Mildew (*B. cinerea*) in Pot

We quantitatively weighed the test sample and dissolved it in 100 μL ofDMSO, diluted to the designed concentration with aqueous solution containing 0.1% Tween 80, and set the blank control without the compound. The solution was sprayed uniformly to the pre-cultured cucumber leaves (BBCH stage 12 102). About 3 h later, when the leaves had dried, the treated cucumbers were inoculated with spores suspension of *Botrytis cinerea* (5–6 × 10^5^ spores/mL), with 4 replicates per treatment. The inoculated test materials were moved to the incubator (RH100%) for 24 h, and then moved to the incubator with a temperature of 18–25 °C, light of 2000 lux, and humidity of 80–90%. The growth status of cucumber seedlings was observed every day. After 5–7 days, when the blank control was fully infected, the incidence was investigated, the disease index was calculated according to the grading standard, and the control effect was calculated by comparison with the blank control. The grading standards are described as follow: grade 0, normal, with no disease spot; grade 1, less than 3 disease spots per leaf; grade 3, 4–6 disease spots per leaf; grade 5, 7–10 disease spots; grade 7, 11–20 disease spots, some of which are dense in patches; and grade 9, dense spots accounting for about 1/4 of the leaf area. The disease index was calculated by the following equation: disease index = (∑(number of cucumber in each grade × grade number))/(total number of surveyed cucumber × 9) × 100. The control effect was calculated by the following formula: control effect = (the disease index of blank − the disease index of treatment)/the disease index of blank × 100.

#### 4.5.3. Bioassay of Cucurbits Powder Mildew (*S. fuliginea*) in Pot

We washed fresh spores on cucumber leaves covered with powdery mildew with purified water with a small amount of surfactant (e.g., Tween 80), filtered with double-layer gauze to establish spore suspension, and controlled the spore concentration of about 100,000/mL for standby. The preparation and spraying of the fungicidal compound solutions were the same as described in Section 4.5.2. Then, the prepared spore suspension of *S. fuliginea* was sprayed onto treated precultured cucumbers (BBCH stage 12 102). After being naturally air-dried, all the cucumbers were moved to the greenhouse (26–28 °C) to culture for about 7–10 days. We counted the area of the spots on a leaf and calculated the control effect according to the ratio of the treated spots to the blank control after 7–8 days. The incidence was investigated by the disease index with the following grade standards: grade 0, normal, no diseases on the leaf; grade 1, the disease-spots area accounted for less than 5% of the leaf area; grade 3, the spots area accounted for about 6–10% of the leaf area; and the spots areas of grade 5, grade 7, and grade 9 were about 11–20%, 21–40%, and above 40%, respectively. The control effect was calculated by the formula set out in Section 4.5.2.

#### 4.5.4. Bioassay of Rice Blast (*P. oryzae*) in Pot

The prepared fungicide solutions were evenly sprayed on precultured rice leaves (BBCH stage 13) with 4 repetitions. About 2–3 h later, when the leaves had dried, the spores suspension of rice blast (2–3 × 10^5^ spores/mL) were inoculated on the leaves by spray. After inoculation, the test material was moved to the moisture box (RH100%) for 24 h, then moved to a 26–28 °C, 2000 lux, and 80–90% humidity incubator. The incidence was investigated by the disease index with the following grade standards: grade 0, normal, no diseases on the leaf; grade 1, the disease spot number was less than 5 per seedling and the length of the spots was less than 1 cm; grade 3, the spot number was about 6–10 per seedling and the length of some spots were above 1 cm; grade 5, the spot number was about 11–25 per seedling and some spots were dense in patches, with the spots area accounting for about 10–25% of the leaf area; and for grade 7 and grade 9, the spot number was larger than 26, the spots were dense in patches, and the spots area accounted for about 26–50% of the leaf area and above 50%, respectively. The control effect was calculated by comparing with the blank control, using the formula set out in Section 4.5.2.

### 4.6. Statistical Analysis

The bioassay data were n repeated samples of two independent tests. Data analysis considered spatially repeated measurements of the same treatment target. Biological performance was examined using one-way ANOVA at a significance level of 0.01 and 0.5 (DPS v. 19.1, DPS Inc., Beijing, China).

## Figures and Tables

**Figure 1 molecules-28-00017-f001:**
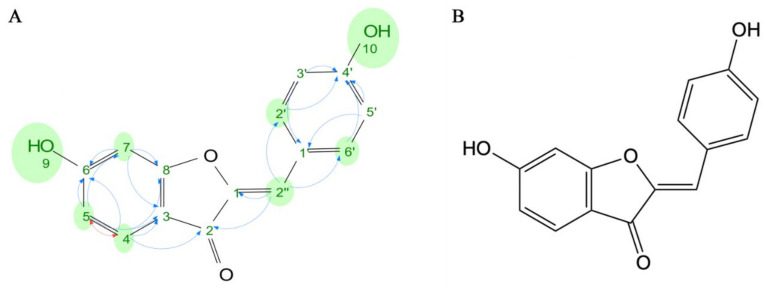
Active substance structure. (**A**) ^1^H–^1^H COSY (H H, red) and HMBC (H C, blue) key correlations of the active compound. (**B**) Chemical structural formula.

**Figure 2 molecules-28-00017-f002:**
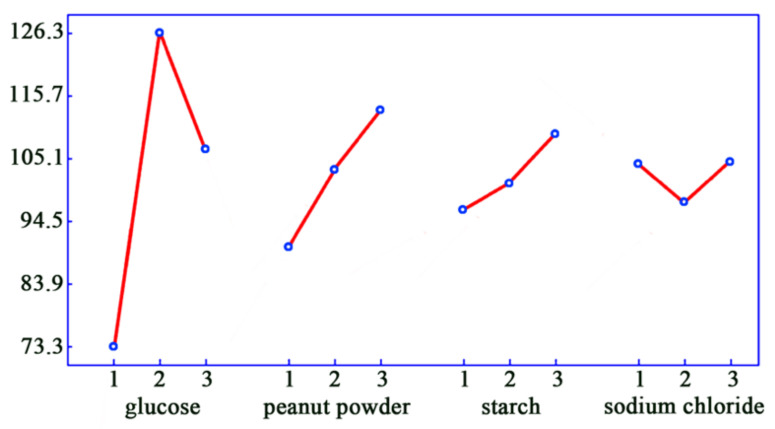
Single-factor intuitvie anlysis chart.

**Figure 3 molecules-28-00017-f003:**
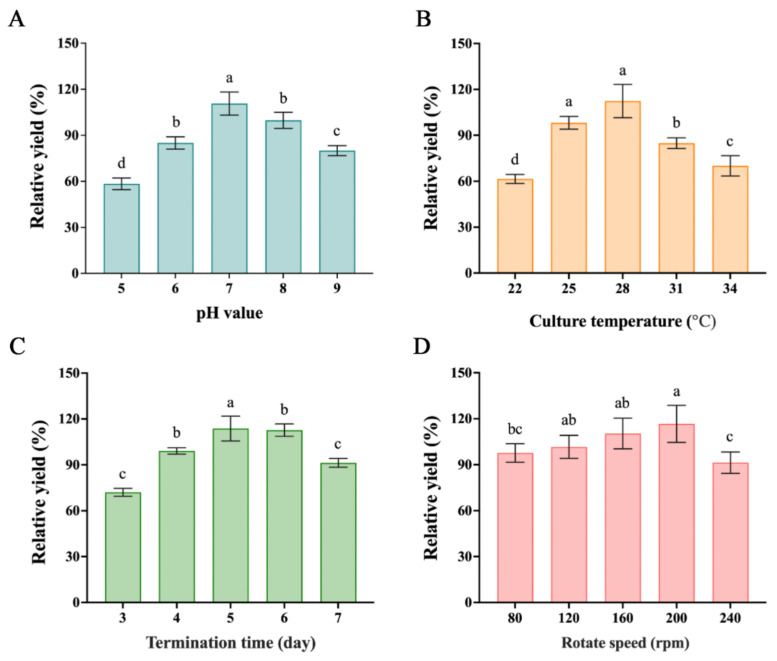
Effects of different fermentation conditions on the yields: (**A**) the effect of pH. relative yields: refer to natural pH; (**B**) the effect of culture temperature—relative yields refer to culture in 28 °C; (**C**) the effect of culture time—relative yields refer to 4 d; (**D**) the effect of shaker rotate speed—relative yields refer to 180 rpm. Data = means ± SD; the letter of alphabet: the same letter means no difference. The data were shown as means ± SD of three independent experiments. *p*-value: a, b, c, and d represent significant differences (*p* < 0.05).

**Figure 4 molecules-28-00017-f004:**
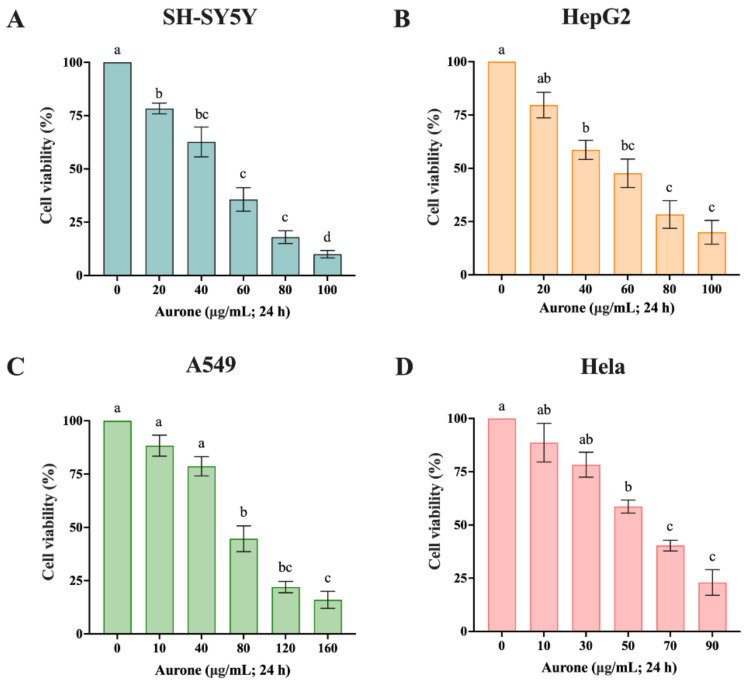
Effects of aurone on the cell viability of different cells. The different lowercase alphabets indicate significant differences between any two sets of treatments (*p* ≤ 0.05). (**A**) SH-SY5Y cells, (**B**) HepG2 cells, (**C**) A549 cells, and (**D**) HeLa cells. The data were shown as means ± SD of three independent experiments. *p*-value: a, b, c, and d represent significant differences (*p* < 0.05).

**Table 1 molecules-28-00017-t001:** Cultural characteristics of strain TXC6.

Medium	Visual Status	Arobic Myclium	Inner Myclium	Soluble Pigment
NA	abundant	gray	coral red	red
Glucose-aginine agar	Rare	cineritious	blue gray	ivory yellow
Salt-starch agar	abundant	pale yellow	yellow	none
Oats powder agar	Rare	light purple	gray	none
PDA	Rare	white clam	gray	black
Yeast-malt agar	abundant	aurora red	litchi white	yellow
Santas agar	abundant	lady gray	shallow tobacco	none
Saccarose-nitrate agar	abundant	cineritious	light megenta	light red

**Table 2 molecules-28-00017-t002:** Physiological and biochemical properties of TXC6.

Biochemistry Characteristics	TXC6	The Use of Carbon Source	TXC6
Glutin hydrolyzation	Positive	L-arabinose	negative
Milk peptonization	Positive	D-glucose	positive
Starch hydrolyzation	Positive	L-rhamnose	negative
Nitrate reduction	Positive	D-fructose	positive
Cellulose usage	Negative	D-xylose	negative
Black pigment	Yield	sucrose	negative
H_2_S	Yield	D-mannitol	positive

**Table 3 molecules-28-00017-t003:** 1H NMR (600 MHz) and 13C NMR (150 MHz) data of in d6-DMSO.

Position	^1^H-NMR	^13^C-NMR
1	-	145.77
2	-	181.31
3	-	113.21
4	7.60 (d, *J* = 8.4 Hz)	125.78
5	6.70 (dd, *J* = 8.4, 2.0 Hz)	112.92
6	-	167.56
7	6.78 (d, *J* = 2.0 Hz)	98.57
8	-	166.21
1′	-	123.09
2′, 6′	7.79–7.84 (m)	133.33
3′, 5′	6.88–6.91 (m)	116.12
4′	-	159.32
2″	6.73 (s)	111.45
9-OH	11.13 (s)	-
10-OH	10.15 (s)	-

**Table 4 molecules-28-00017-t004:** Relative yield of aurones in different carbon and nitrogen sources in single-factor analysis.

Carbon Source	Relative Yield * (%)	Nitrogen Source	Relative Yield (%)
Glucose	100	Soybean powder	100
Fructose	81.25 ± 0.75	Peanut powder	108.30 ± 1.73
Maltose	83.75 ± 1.02	Yeast extract powder	105.65 ± 0.98
Sucrose	27.50 ± 0.33	Peptone	102.65 ± 1.25
Starch	93.75 ± 2.18	Potassium nitrate	68.35 ± 0.67

* Relative yield refers to the ratio of yield when glucose is the only carbon source to that of other carbon sources, or the ratio of yield when soybean powder is the only nitrogen source. Data = mean ± SD, *n* = 4.

**Table 5 molecules-28-00017-t005:** L9(3^4^) orthogonal design.

Factor Level	Glucose (A)/g·L^−1^	Peanut Powder (B)/g·L^−1^	Starch (C)/g·L^−1^	Sodium Chloride (D)/g·L^−1^
1	2.0	1.0	0.5	0.5
2	4.0	2.0	1.0	1
3	6.0	3.0	2.0	2

Data = mean ± SD, *n* = 3.

**Table 6 molecules-28-00017-t006:** The result of L9(3^4^) orthogonal design.

Experimental Number	Factors and Levels				Yield/mg·L^−1^
	A	B	C	D	
1	1	1	1	1	57.67 ± 4.51
2	1	2	2	2	68.33 ± 3.21
3	1	3	3	3	94.00 ± 4.58
4	2	1	2	3	115.33 ± 4.93
5	2	2	3	1	136.67 ± 4.72
6	2	3	1	2	127.00 ± 9.84
7	3	1	3	2	97.33 ± 2.08
8	3	2	1	3	104.33 ± 7.09
9	3	3	2	1	118.33 ± 3.05
R	53	23	13	7	
R’	47.73	20.72	11.71	6.03	
F value	224.20	41.54	13.68	4.87	
*p* value	0	0	0.0002	0.0204	

Data = mean ± SD, *n* = 3.

**Table 7 molecules-28-00017-t007:** The control effect of compounds against different disease in pot test.

Treatment Chemicals	Concentration/mg·L^−1^	Pathogens and Control Effect/%					
		*P. orizae*		*B. cinerea*		*S. fuliginea*	
		Disease Index	Control Effect	Disease Index	Control Effect	Disease Index	Control Effect
CK	0	55.80 ± 1.89 ^a^	/	78.72 ± 2.03 ^a^	/	79.12 ± 2.10 ^a^	/
Purified aurone	25	27.42 ± 4.25 ^b^	51.01 ± 6.26 ^e^	42.07 ± 3.01 ^b^	46.57 ± 3.66 ^e^	53.09 ± 3.39 ^b^	32.91 ± 4.09 ^f^
	50	16.08 ± 0.86 ^d^	71.18 ± 1.17 ^c^	22.67 ± 1.90 ^d^	71.20 ± 2.38 ^c^	23.47 ± 1.89 ^d^	70.35 ± 2.25 ^d^
	100	1.02 ± 0.38 ^f^	98.16 ± 0.71 ^a^	1.20 ± 0.75 ^f^	98.48 ± 0.96 ^a^	5.61 ± 1.36 ^f^	92.91 ± 1.68 ^b^
Crude extra ointment	25	20.57 ± 1.07 ^c^	63.15 ± 1.50 ^d^	29.30 ± 2.00 ^c^	62.79 ± 2.39 ^d^	34.08 ± 2.95 ^c^	56.92 ± 3.76 ^e^
	50	10.03 ± 1.05 ^e^	82.04 ± 1.76 ^b^	14.66 ± 1.12 ^e^	81.37 ± 1.51 ^b^	16.82 ± 1.97 ^e^	78.74 ± 2.40 ^c^
	100	0.00 ± 0.00 ^f^	100.00 ± 0.00 ^a^	0.00 ± 0.00 ^f^	100.00 ± 0.00 ^a^	0.00 ± 0.00 ^g^	100.00 ± 0.00 ^a^
Tricyclazole	20	0.38 ± 0.28 ^f^	99.32 ± 0.49 ^a^	/	/	/	/
Carbendazim	500	/	/	0.39 ± 0.24 ^f^	99.50 ± 0.31 ^a^	/	/
Difenoconazole	100	/	/	/	/	0.45 ± 0.44 ^g^	99.43 ± 0.56 ^a^

The letters a–g indicated significant difference (*p* < 0.05). Data = mean ± SD, *n* = 8.

## Data Availability

Not applicable.

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
