# Peer review of "A New Method of Preparing Aurone by Marine Actinomycetes and Its Potential Application in Agricultural Fungicides"

_molecules, 2022, doi:10.3390/molecules28010017_

Round 1
Reviewer 1 Report
The work is interesting in a scientific sense, but the text needs to be improved.
The sentences are completely confused, unclear, especially in the introduction and results. Words like "tedious" are inappropriate for scientific work. English language and style is very bad.
Words like "tedious" are inappropriate for scientific work. It is also inappropriate to write :"we thought".
If you worked with A549 cell line (a widely used human lung adenocarcinoma cell line that was derived from a primary lung tumor). Hep G2 cell lines (isolated from a hepatocellular carcinoma) why do you call them canner cells???
Use italics in the names of all types of organisms you list.
In the results under 2.2., the sentences are completely confused, it is not possible to follow the context or the obtained results. Expand the sentences with explanations.
When you say strong inhibition, you have to specify what ... strong inhibithion of growth (results 2.4.)
In the results under number 2.5, it is necessary to better explain the control effect. Did you mean the fungicidal effect?
Author Response
Dear Reviewer:
Thank you for your review. We have made modifications according to your comments. We replace cancer cells with tumor cells. Results 2.2, 2.4 and 2.5 were modified accordingly.. In addition, we invited native English speaking colleagues to revise the article. Thank you again.
Best Regards.
Yours Sincerely
Yang Zhang
Reviewer 2 Report
In the manuscript “A new method of preparing aurone by marine actinomycetes and its potential application in agricultural fungicides” it has been done good job and it should be accepted with minor revision. The authors should do extensive editing of English language and typing errors. For instance, in line 72 they should write “combined” instead of “combinend”.
Line 72
combined instead of combinend
Author Response
Dear Reviewer:
Thank you for your review. We have made modifications according to your comments. And we invited native English speaking colleagues to revise the article. Thank you again.
Best Regards.
Yours Sincerely
Yang Zhang
Reviewer 3 Report
In this manuscript, a strain of marine actinomycetes was isolated from intertidal zone and identified as Strep-tomyces cinereoruber, and the fermentation parameters were determined by the method of fermentation optimization.The bioassay in greenhouse showed that the aurone had high fungicidal activities againstagainst three plant fungi. I think the manuscrip can be accepted after minor resision.
1. Page 3, "calcd for C15H10O4, 254.24)." ,please give four decimal places.
2. Page 3, "Together with 13C NMR, DEPT, COSY", please make a further description.
3. Page 4, "Figure 1.", give a clear structure of the active compound.
Author Response
Dear Reviewer:
Thank you for your review. We have made modifications according to your comments. In the first suggestion, we've added 4 decimals. In the second comment, we added an interpretation of the results. In the third comment, we added the structure of the compound in Figure 1. In addition, we invited native English speaking colleagues to revise the article. Thank you again.
Best Regards.
Yours Sincerely
Yang Zhang